# Double-Edged Sword of Vitamin D3 Effects on Primary Neuronal Cultures in Hypoxic States

**DOI:** 10.3390/ijms22115417

**Published:** 2021-05-21

**Authors:** Maria Loginova, Tatiana Mishchenko, Maria Savyuk, Svetlana Guseva, Maria Gavrish, Mikhail Krivonosov, Mikhail Ivanchenko, Julia Fedotova, Maria Vedunova

**Affiliations:** 1Department of Neurotechnology, Institute of Biology and Biomedicine, Lobachevsky State University of Nizhny Novgorod, 23 Gagarin Ave., 603022 Nizhny Novgorod, Russia; pandaagron@yandex.ru (M.L.); saHarnova87@mail.ru (T.M.); mary.savyuk@bk.ru (M.S.); mary.kasatkina2011@yandex.ru (S.G.); mary_gavrish@mail.ru (M.G.); julia.fedotova@mail.ru (J.F.); 2Department of Applied Mathematics, Institute of Information Technologies, Mathematics and Mechanics (ITMM), Lobachevsky State University of Nizhny Novgorod, 23 Gagarin Ave., 603022 Nizhny Novgorod, Russia; mike_live@mail.ru (M.K.); ivanchenko.mv@gmail.com (M.I.); 3Laboratory of Neuroendocrinology, I.P. Pavlov Institute of Physiology, Russian Academy of Sciences, 6 Emb. Makarova, 199034 St. Petersburg, Russia

**Keywords:** vitamin D3, cholecalciferol, hypoxia, neuroprotection, primary neuronal cultures, functional neural network activity

## Abstract

The use of vitamin D3 along with traditional therapy opens up new prospects for increasing the adaptive capacity of nerve cells to the effects of a wide range of stress factors, including hypoxia-ischemic processes. However, questions about prophylactic and therapeutic doses of vitamin D3 remain controversial. The purpose of our study was to analyze the effects of vitamin D3 at different concentrations on morpho-functional characteristics of neuron–glial networks in hypoxia modeling in vitro. We showed that a single administration of vitamin D3 at a high concentration (1 µM) in a normal state has no significant effect on the cell viability of primary neuronal cultures; however, it has a pronounced modulatory effect on the functional calcium activity of neuron–glial networks and causes destruction of the network response. Under hypoxia, the use of vitamin D3 (1 µM) leads to total cell death of primary neuronal cultures and complete negation of functional neural network activity. In contrast, application of lower concentrations of vitamin D3 (0.01 µM and 0.1 µM) caused a pronounced dose-dependent neuroprotective effect during the studied post-hypoxic period. While the use of vitamin D3 at a concentration of 0.1 µM maintained cell viability, preventive administration of 0.01 µM not only partially preserved the morphological integrity of primary neuronal cells but also maintained the functional structure and activity of neuron–glial networks in cultures. Possible molecular mechanisms of neuroprotective action of vitamin D3 can be associated with the increased expression level of transcription factor HIF-1α and maintaining the relationship between the levels of BDNF and TrkB expression in cells of primary neuronal cultures.

## 1. Introduction

Hypoxia is one of the key factors determining the pathogenesis of a wide range of diseases of the central nervous system, including ischemic stroke, neuro-oncology, neurodegenerative diseases (e.g., Alzheimer’s disease, Parkinson’s disease, Huntington’s disease, Friedreich’s ataxia, epilepsy, age-related memory impairment, multiple sclerosis). The brain is the main target organ for the destructive effects of hypoxia due to its need to consume large amounts of oxygen to maintain its metabolic and functional activity, and a limited set of antioxidant enzymes and compensatory capabilities [1,2,3]. Under reduced oxygen content conditions, pathological cascades of reactions aimed at uncoupling oxidative phosphorylation, violations of cellular energy metabolism, and the activation of free radical processes are launched. The consequences of these processes are mitochondrial dysfunction, impaired synaptic transmission, stimulation of inflammatory and apoptotic reactions leading to cell death and destruction of functionally important elements of neuron–glial networks in the brain [3,4].

Researchers’ attention has been focused on studying the pathogenetic aspects of hypoxia and the search for therapeutic strategies aimed at activating the adaptive capabilities of the nervous system to hypoxic damage. With regard to this issue, the use of vitamin D seems to be one of the most attractive therapeutic targets. 

Vitamin D3 is a steroid hormone that regulates calcium and phosphorus homeostasis in the organism. The vitamin D3 action is mediated by binding to the VDR protein. This complex is transported to the cellular nuclei and activates genes that maintain calcium homeostasis in the blood [5,6]. Vitamin D is also a powerful neurosteroid that is directly involved in neurogenesis, regulation of behavioral and neuroendocrine functions in the CNS, and neuroprotection. 

Vitamin D3 deficiency leads to changes in glutamate transporter regulation and the balance of excitatory and inhibitory neurons, which is associated with a wide range of neuropsychiatric disorders and neurodegenerative diseases [7,8]. Several studies have shown that children born with diagnosed hypoxic-ischemic encephalopathy tend to have vitamin D3 deficiency [9,10]. The latest experimental studies have indicated that vitamin D, which penetrates the blood–brain barrier, has a pronounced neuroprotective effect in the development of hypoxia-ischemic processes. For instance, the use of a neonatal ischemia rat model (P3–5) showed that intraperitoneal and enteral administration of vitamin D contributes to decreasing the spread of ischemic stroke and activates anti-inflammatory mechanisms in the brain tissue, inducing activation of IL-4 synthesis and suppressing the expression of pro-inflammatory cytokines IL-6 by astrocytes [10,11,12]. Moreover, intraperitoneal administration of calcitriol to male rats after global cerebral ischemia reduced cerebral edema and contributed to the maintenance of neurological function. At the same time, the use of vitamin D3 led to the activation of ERK ½ kinase, thereby reducing neuronal apoptosis activity [13].

The potential use of vitamin D as a powerful therapeutic target in hypoxic damage dictates the need for more detailed studies of the mechanisms of the vitamin D action on the functional activity of the nervous system’s cells in both normal conditions and under stress.

The current work aimed to study the effects of vitamin D3 at various concentrations on the morpho-functional characteristics of neuron–glial networks in hypoxia modeled in vitro.

## 2. Results

The starting point of the study was estimating the effect of vitamin D3 at various concentrations on the viability and functional activity of cells in primary neuronal cultures under normal conditions. 

The viability analysis revealed that the use of all studied concentrations of vitamin D3 does not have a pronounced cytotoxic effect on the primary culture cells. One day after the vitamin D3 addition, the number of necrotic and apoptotic cells in the experimental groups did not differ from the sham values (Appendix A). Notably, the use of vitamin D3 at a concentration of 0.01 µM decreased the ratio of necrotic to apoptotic cells compared to the “Sham” group. On day 7 after vitamin D3 addition, the number of viable cells in the experimental groups did not differ from the sham and control values (Table 1).

Despite the absence of significant morphological changes, the use of vitamin D3 led to modulation of the functional calcium activity of primary neuronal cultures. The addition of vitamin D3 at a concentration of 1 µM significantly decreased the number of cells exhibiting calcium activity (5.31 ± 0.66%), accompanied by an increase in the duration (14.99 ± 0.68 s) and a decrease in the frequency (0.25 ± 0.05 osc/min) of Ca^2+^ oscillations. A tendency towards a decrease in the number of functionally active cells in the culture (35.41 ± 10.92%), accompanied by a significant increase in the duration of Ca^2+^ oscillations (9.91 ± 0.45 s) was shown for the “vitamin D3 0.1 µM” group. The use of vitamin D3 at a concentration of 0.01 µM did not significantly affect the spontaneous calcium activity of primary neuronal cultures. The main parameters of calcium activity in the “vitamin D3 0.01 µM” group were comparable to that in the sham values (Figure 1, Appendix A).

Analysis of connectivity characteristics between elements in the neuron–glial network showed that the use of a high concentration of vitamin D3 (1 μM) leads to pronounced disruption of the network’s functional structure (Figure 2). Application of vitamin D3 at a concentration of 0.1 μM caused a significant decrease in the rate of calcium signal delay (20.66 (19.46; 21.67)) compared to the control values (18.72 (12.60; 20.45)), and the remaining parameters were unchanged relative to the “Sham” group.

The construction of correlation network graphs showed that the use of a high concentration of vitamin D3 (1 μM) caused almost complete destruction of long and short connections, which indicates a pronounced functional degradation of the neuron–glial network (Appendix A). Similar to control cultures, the cultures that received the lower concentrations of vitamin D3 are represented by highly correlated neuron–glial networks with a large number of both short and long-range connections. The graphs of correlation dependence between a pair of cells and the distance between them demonstrate the correlations of Ca^2+^ oscillations between cells and their level. As seen in the graphs, the use of a high concentration of vitamin D3 destroyed all highly correlated connections.

Next, we studied the effects of vitamin D3 of different concentrations on nerve cell’s adaptation to hypoxic damage. 

The viability assay revealed that hypoxia leads to a decrease in the cell viability of primary neuronal cultures (Table 2). Besides, there is an increase in the number of cells dying, both by necrosis and apoptosis pathways (Appendix A). The use of low concentrations of vitamin D3 contributes to maintaining the nerve cell’s viability. One day after hypoxia modeling, the number of necrotic and apoptotic cells was reduced compared to the “Hypoxia” group and did not differ from the sham values (Appendix A). Interestingly, the use of vitamin D3 at a concentration of 0.01 µM contributes to maintaining the ratio of necrotic to apoptotic cells within the sham values. It can be assumed that the anti-hypoxic effect of vitamin D3 is implemented through modulating intracellular processes, activating a more energy-consuming but safe-for-cell elimination system in culture. In the late post-hypoxic period, the number of viable cells in the “Hypoxia + vitamin D3 0.01 µM” and “Hypoxia + vitamin D3 0.1 µM” groups was 89.34 ± 0.99% and 86.79 ± 0.86%, respectively, which significantly differed from the values of the “Hypoxia” group (80.39 ± 2.17%) (Table 2). The pronounced neuroprotective effect remained throughout the observation period.

Interestingly, the use of vitamin D3 at a concentration of 1 µM in hypoxic state, not only had no neuroprotective effect but also led to total cell death in primary neuronal cultures. Already on the first day after hypoxia modeling, the number of necrotic and apoptotic cells exceeded the values of the “Sham” group by 15.4 times and 6 times, respectively (Appendix A). On the third day of the post-hypoxic period, the number of viable cells in the “Hypoxia + vitamin D3 1 µM” group was 8.89 ± 2.6%, and by day 7, viable cells were not recorded in the culture (Table 2).

Analysis of the functional calcium activity also confirmed the pronounced neuroprotective effect of low concentrations of vitamin D3 (Figure 3). The use of a concentration of 0.01 µM maintained the number of functionally active cells in culture. On day 7 of the post-hypoxic period, the number of cells exhibiting Ca^2+^ activity in the “Hypoxia + vitamin D3 0.01 µM” group was 44.63 ± 7.48%, which significantly differed from the values of the “Hypoxia” (24.89 ± 4.74%) and “Hypoxia + solvent” (24.47 ± 6.08%) groups. The frequency of Ca^2+^ oscillations in the “Hypoxia + vitamin D3 0.01 µM” (0.99 ± 0.16 osc/min) also exceeded the control values (“Hypoxia” 0.59 ± 0.11 osc/min, “Hypoxia + solvent” 0.62 ± 0.23 osc/min) (Appendix A).

The use of vitamin D3 at a concentration of 0.1 µM contributed to maintaining the viability of primary neuronal cultures in the post-hypoxic period but did not have a protective effect on the functional activity of cells. No significant changes were found for any of the studied parameters in comparison with the controls (Figure 3) (Appendix A).

Preventive use of vitamin D3 at a concentration of 1 µM leads to the complete destruction of neuron–glial networks and negation of their functional activity.

Assessment of the network characteristics on day 3 after hypoxia modeling did not reveal significant differences in the functional structure of the neuron–glial networks in the control and experimental groups (Appendix A).

However, in a later post-hypoxic period, significant violations of the functional structure of the neuron–glial networks were determined (Figure 4). A decrease in all network characteristics was shown for the “Hypoxia” group: the average level of correlation of all cells decreased from 0.48 (0.44; 0.68) to 0.27 (0.19; 0.34); the average level of correlation of adjacent cells decreased from 0.66 (0.59; 0.78) to 0.45 (0.38; 0.56); the average number of connections for each individual cell decreased from 377.34 (341.48; 416.22) connections to 180.73 (64.85; 251.16) connections; the percentage of existing connections of the maximum possible number of connections in culture decreased from 80.44 (69.88; 96.51) to 34.89 (13.83; 47.64), and the average rate of calcium signal delay between cell pairs decreased from 18.72 (12.60; 24.93) to 12.45 (9.34; 19.17).

There was a tendency for network parameters to increase upon vitamin D3 application compared to the control values. When the signal transmission rate remained unchanged, in the group “Hypoxia + vitamin D3 0.01 μM”, an increase in the median value was found in terms of the level of correlation of all cells (Hypoxia + vitamin D3 0.01 μM: 0.47 (0.13; 0.56), Hypoxia: 0.27 (0.20; 0.34)); correlations of adjacent cells (Hypoxia + vitamin D3 0.01 μM: 0.62 (0.40; 0.66), Hypoxia: 0.45 (0.38; 0.56)); the average number of connections per cell (Hypoxia + vitamin D3 0.01 μM: 353.35 (17.57; 389.36), Hypoxia: 180.73 (64.85; 251.16)), and the percentage of correlated connections from the total number of possible connections (Hypoxia + vitamin D3 0.01 μM: 367.75 (3.47; 81.50), Hypoxia: 34.89 (13.83; 47.64)) (Figure 4a–d). Such changes in the studied parameters of network activity suggest that the vitamin D3 application has a mild neuroprotective effect on the functional structure of the neuron–glial network.

Analysis of the correlation network graphs revealed that hypoxic damage leads to a significant decrease in functional connections between the neuron–glial network elements (Appendix A). The use of vitamin D3 contributes to maintenance of the functional interactions between cells. The most pronounced effect was shown for the “Hypoxia + vitamin D3 0.01 µM” group. Similar can be stated based on the graphs of the dependence of calcium oscillations correlation level between a pair of cells on the distance between them. The correlation level in the “Hypoxia” group decreased, and the point cloud shifted to a lower value of correlation. This suggests that hypoxia led to the degradation of the neural-glial network connections and decreases the calcium signal communication level in the preserved connections. This effect was not observed upon vitamin D3 application; the correlation level for some connections remained at a high level.

The expression level of the neurotrophic factor BDNF and the second subunit of transcription factor HIF-1 (HIF-1α) was studied in order to identify possible molecular mechanisms of the neuroprotective effect of vitamin D3 in hypoxic damage (Figure 5a,b). HIF-1α is a key subunit whose expression depends on the oxygen concentration in the cell and is the main cell response to hypoxia [14,15]. The analysis revealed that hypoxia leads to a significant decrease in the expression level of both HIF-1α and BDNF. Compared to the sham values, the amount of HIF-1α mRNA and BDNF mRNA in the “Hypoxia” group decreased by 0.73 times and 0.47 times, respectively. Furthermore, in the “Hypoxia” group, a tendency to a decrease in the expression level of vitamin D receptor (VDR) was also noted (Figure 5c). At the same time, hypoxia did not affect the expression of a high-affinity receptor for BDNF–tropomyosin receptor kinase B (TrkB) (Figure 5d).

Previous studies have shown that the level of HIF-1α expression increases sharply in chronic hypoxia and returns to normal levels only after 3 weeks of the post-hypoxic period [16]. In acute hypoxic and hypoxia-ischemia conditions, the level of HIF-1α expression was upregulated at 4 h, peaked at 8 h, and declined at 24 h after the modeled stress in rats [17]. Based on our data, it could be assumed that a day after hypoxia modeling in vitro, the factors suppressing HIF1α expression were activated, followed by the depletion of accumulated HIF1α for damage repair and a decrease in the amount of HIF1α mRNA in the cell in the early post-hypoxic period. The hypoxic influence also dramatically decreased the level of Cav 1.2 subunit of L-type calcium channels (Appendix A), which can be considered a compensatory effect aimed at suppressing the intracellular Ca^2+^ level and the development of excitotoxicity. However, more prolonged suppression of L-type calcium channels expression can serve as a prerequisite for impaired Ca^2+^ ions conductivity and further development of the violations of functional activity in neuron–glial networks observed in the distant post-hypoxic period.

The use of vitamin D3 led to an increase in the level of HIF-1α expression. The most pronounced effect was noted for the “Hypoxia + vitamin D3 1 μM” group, in which the HIF-1α expression level exceeded the values of the control group by 1.23 times. Besides, the level of VDR expression in this experimental group was comparable to the sham values. On the other hand, in parallel with a tendency to maintain the level of BDNF expression there was a significant decrease in TrkB receptors expression, which indicates only short-term activation of adaptive effects in the early post-hypoxic period. Moreover, the use of vitamin D3 at a concentration of 1 μM dramatically decreased the expression level of both Cav 1.2 and Cav 1.3 subunits of L-type calcium channels by 10 times and 4 times, respectively (Appendix A), which also serves as one of the causes of complete destruction of functional neuron–glial networks by day 3 of the post-hypoxic period.

The use of other concentrations of vitamin D3 maintained the level of HIF-1α. The level of BDNF expression in the “Hypoxia + vitamin D3 0.01 μM” group was reduced compared to the sham values, but significantly exceeded the values of the “Hypoxia” group (by 1.7 times). Thus, the use of vitamin D3 at a concentration of 0.1 μM leads to a decrease in the level of BDNF expression and an increase in the expression of TrkB receptors, which indicates the potential activation of adaptive compensatory mechanisms maintaining the neuroprotective BDNF functions. No significant alterations in the level of VDR as well as Cav 1.2 and Cav 1.3 expression in both experimental groups were found compared to the “Hypoxia” group. Thus, it can be assumed that the neuroprotective effect of vitamin D3, which lasted for 7 days after the modeled hypoxia, is mediated by the genetic regulation of the HIF-1α gene expression and possible maintenance of the relationship between the levels of BDNF and TrkB expression in cells of primary neuronal cultures.

## 3. Discussion

Experimental and clinical biomedical studies performed in recent decades clearly confirm that vitamin D effects are beyond the regulation of calcium and phosphorus homeostasis. Vitamin D sufficiency is vital for the most important physiological functions in an organism [18,19,20]. As for the nervous system, being a neurosteroid hormone, vitamin D significantly affects the synthesis of neurotrophins and neurotransmitters, maintains intracellular calcium homeostasis and has a neuroprotective effect when exposed to a number of stress factors [21,22,23,24].

The steady increase in vitamin D deficiency in the global population poses a significant threat to public health and risks becoming a global metabolic pandemic of the 21st century [25]. The status of vitamin D in the organism depends on sun exposure, diet, intake of its supplements, lifestyle and genetic factors [26,27]. A low level of vitamin D is associated with the development of a wide range of acute and chronic diseases, including calcium metabolism disorders [28], neuropsychiatric disorders and neurodegenerative diseases [29,30], allergic and autoimmune reactions [31,32], endocrine disorders (including fertility) [33,34], some types of cancer [35], type 2 diabetes [36], cardiovascular [37] and infectious diseases [38]. Recent studies have also shown the correlation between the level of vitamin D and the severity of affective disorders in patients in the peri- and postmenopausal periods [39,40,41]. Vitamin D deficiency significantly complicates the condition of patients with ischemic stroke and increases the risk of cognitive impairment and dementia [12,42,43,44]. 

Thus, the replenishment of vitamin D along with conventional treatments is becoming increasingly relevant, opening up new possibilities for clinical application. Nevertheless, the prophylactic and therapeutic doses of vitamin D, its pharmacokinetics and pharmacodynamics features, and the modes and frequency of its administration remain controversial. This is primarily due to the fact that vitamin D intoxication accompanied by a high level of calcium and phosphorus can cause hypercalcemia, kidney damage, vascular calcification and activation of other pathological processes, including those in the nervous tissue [45].

Our studies have shown that a single administration of vitamin D3 at high concentrations in normal conditions does not significantly affect the cell viability in primary neuronal cultures; however, it exhibits a pronounced modulating effect on the functional calcium activity of neuronal-glial networks. The observed effect is characterized by a decrease in the number of metabolically active cells, an increase in the duration and a decrease in the frequency of Ca^2+^ oscillations. Analysis of network characteristics revealed almost complete destruction of long and short connections in the neuron–glial network, and the destruction of all highly correlated connections, which indicates pronounced functional destruction of the network response. In hypoxic conditions, aggravation of the toxic effects of vitamin D3 at high concentration was observed. The preventive administration of vitamin D3 (1 µM) led to the total death of cells in primary neuronal cultures and complete inhibition of functional neural network activity. It is known that vitamin D3 can activate glutamatergic transmission via allosteric regulation of NMDA and AMPA receptors [46]. Therefore, it can be assumed that the hyperactivation of excitatory receptors, mediated by high doses of vitamin D3, can cause an excessive intake of Ca^2+^ ions into the cell, thereby increasing the development of excitotoxicity by the hypoxic damage. The active entry of Ca^2+^ is accompanied by the activation of a number of enzymes (phospholipases, endonucleases, calpains) that have a destructive effect on cytosolic structures, ultimately leading to the triggering of apoptosis [46,47].

On the contrary, the preventive vitamin D3 administration at lower concentrations (0.01 µM and 0.1 µM), had a pronounced neuroprotective effect, which persisted throughout the studied post-hypoxic period. The observed neuroprotective effect was dose-dependent. The use of a concentration of 0.1 µM contributed to the maintenance of cell viability. In comparison, the concentration of 0.01 µM partially preserved the cell’s morphological integrity and maintained the activity of neuron–glial networks at a certain functional level. Thus, the vitamin D3 administration increased the proportion of cells showing Ca^2+^ activity and the frequency of Ca^2+^ oscillations in primary neuronal cultures compared to the control values. In addition, the use of vitamin D3 contributed to the partial preservation of the neuron–glial network functional structure, characterized by the maintenance of functional intercellular connections and the high correlation level between cells.

High concentrations of vitamin D3 are associated with increased hypoxia-induced excitotoxic effects; on the contrary, low concentrations can affect the adaptive mechanisms of nerve cells. The revealed neuroprotective effect of low concentrations of vitamin D3 can be mediated in several pathways. For example, vitamin D3 is able to suppress Ca^2+^ levels by stimulating the expression of calcium-binding proteins (parvalbumin and calbindins-D9k and -D28k) and inhibiting the expression of L-type Ca^2+^ channels in hippocampal cells [48,49]. Besides, vitamin D3’s antioxidant properties are associated with its active participation in the regulation of γ-glutamyltranspeptidase, the key enzyme of glutathione metabolism, thereby contributing to reducing the level of reactive oxygen species, in particular hydrogen peroxide, in a number of stress factors including ischemia [24,50]. Recent studies have also showed that the neuroprotective effect of vitamin D in a hypoxic state can be implemented through downregulating dual oxidase 1 (DUOX1) via the NF-kB signaling pathway, and therefore attenuate the production of a large number of reactive oxygen species induced by hypoxia [51]. Calcitriol, the main biologically active form of vitamin D3, can activate the intracellular signaling cascade MEK/ERK; the launch of this kinase pathway leads to activation of transcription factor CREB and blockade of apoptosis [52]. Besides, the induction of the synthesis of a number of neurotrophic factors (e.g., BDNF, NT-3, NT-4, NGF) and the regulation of the expression of the transcription factor HIF-1α could be one of the possible neuroprotective mechanisms of vitamin D3 [53,54,55]. In the current study, we showed that in parallel with a decrease in L-type calcium channel expression, the use of vitamin D3 increases the expression level of HIF-1α and maintains the relationship between the levels of BDNF and TrkB expression in neuronal cells, which suggests the possible short-term activation of adaptive effects in the early post-hypoxic period.

## 4. Materials and Methods

### 4.1. Ethics Statement

The animals were housed in a certified SPF vivarium of Lobachevsky State University of Nizhny Novgorod. Experiments were carried out in accordance with Act 708n (23 August 2010) of the Ministry of Health of the Russian Federation, which states the rules of laboratory practice for the care and use of laboratory animals, and Council Directive 2010/63 EU of the European Parliament (22 September 2010) on the protection of animals used for scientific purposes and were approved by the Bioethics Committee of Lobachevsky State University of Nizhny Novgorod (protocol No46 from 19 October 2020). C57BL/6J mice were killed by cervical vertebra dislocation, and their embryos were then surgically removed and sacrificed by decapitation. 

### 4.2. Experimental Design

Vitamin D3 (cholecalciferol, Sigma-Aldrich, Steinheim Germany) was dissolved in 1 mL of 95% ethanol. The obtained aliquots were then stored at 80 °C. The following concentrations of vitamin D3 were used: 0.01 µM, 0.1 µM and 1 µM.

For cytotoxicity analysis, vitamin D3 was added to the culture medium on day 14 of the primary neuronal culture’s development (DIV). The “Solvent” group consisted of primary neuronal cultures with the addition of solvent (0.1% ethanol). 

To study the neuroprotective effects of vitamin D3, the tested substance was added to the culture medium 20 min before hypoxia modeling, during hypoxia and immediately after reoxygenation. The “Solvent” group consisted of primary neuronal cultures subjected to hypoxia with the addition of solvent (95% ethanol).

### 4.3. Isolation of the Primary Neuronal Cultures

Primary neuronal cells were obtained from the cerebral cortex and hippocampus of C57BL/6J murine embryos (day 18 of gestation) according to previously developed protocols [56]. Dissociation of cells was achieved by treating the brain tissue with a 0.25% trypsin solution (Thermo Fisher, Waltham, MA, USA). The suspension of dissociated cells was centrifuged at 1000 rpm for 3 min. The cells were cultured in Neurobasal^TM^ medium (Thermo Fisher, Waltham, MA, USA) supplemented with 5% fetal calf serum (PanEco, Moscow, Russia), 0.5 mM L-glutamine and 2% B27 (Thermo Fisher, Waltham, MA, USA). The initial cell density in the culture was 7000–9000 cells/cm^2^. The culture’s viability was maintained in a CO_2_-incubator (BINDER GmbH, Tuttlingen, Germany).

### 4.4. In Vitro Hypoxia Model 

The in vitro hypoxia model was performed on day 14 of cultivation as described in [57], by replacing the culture medium with a medium with a low oxygen content for 10 min, followed by a reverse replacement of the growth medium. The hypoxic medium was created by passing argon gas through the Neurobasal^TM^ medium in a sealed chamber at a pressure of 1–1.5 MPa. According to the Winkler test, the oxygen concentration in a medium decreased from 3.26 mL/L (normoxia) to 0.37 mL/L (hypoxia). The cultures from the “Sham” group were subjected to total replacement of the culture medium by a complete growth medium with normal oxygen content.

### 4.5. Cell Viability Analysis

The viability assay was conducted by staining primary neuronal cultures with fluorescent dyes, propidium iodide (Sigma-Aldrich, Steinheim Germany) (dead cell nuclei) and bisbenzimide (Sigma-Aldrich, Steinheim, Germany) (total number of cells in culture). Propidium iodide and bisbenzimide at concentrations of 5 μg/mL and 1 μg/mL, respectively, were added to the culture medium 30 min before viability measurements. The stained cells were visualized using a ZEISS Observer A1 inverted fluorescence microscope (Carl Zeiss, Oberkochen, Germany). The proportion of dead cells was calculated as the ratio of propidium iodide positive cells to bisbenzimide positive cells. Each experimental and control groups included five cultures; 10 fields of view were analyzed for each culture. Three independent biological experiments were performed.

In addition, the type of cell death was determined using Tali^TM^ Apoptosis Kit (Thermo Fisher, Waltham, MA, USA) in accordance with the manufacturer’s instructions. In the culture, AnnexinV positive cells were identified as apoptotic, while propidium iodide positive cells were considered to be dying by the necrosis pathway.

### 4.6. Calcium Imaging

The functional metabolic activity of cells in primary neuronal cultures was studied according to the calcium imaging technique. Spontaneous calcium activity of cells was recorded using a calcium-sensitive dye Oregon Green 488 BAPTA-1 AM (OGB-1) (0.4 mM, Thermo Fisher, Waltham, MA, USA) and Zeiss 800 LSM confocal laser scanning microscope (Carl Zeiss, Oberkochen, Germany). OGB-1 was excited at a wavelength of 488 nm; fluorescence emission was recorded in the rage of 500–530 nm. The resolution of the obtained image was 512 × 512 pixels, the size of the field of view was 420 × 420 µm, and the image recording frequency was 2 Hz. Detection and analysis of calcium oscillations were performed using the Astroscanner program [56,58]. The following parameters were assessed: percentage of cells exhibiting Ca^2+^ activity (%), the duration (time from the beginning to the end of an oscillation, s) and frequency (average number of oscillations per min) of Ca^2+^ oscillations.

### 4.7. Analysis of Network Characteristics of Primary Neuronal Cultures Activity

To analyze the network characteristics, we used a previously developed algorithm for calcium activity detection in neuronal cells [59,60]. The algorithm is based on fluctuations in the intracellular calcium level. This allows the representation of the neuron–glial network in the form of an oriented graph with nodes corresponding to individual cells; the edges connect the corresponding nodes and indicate a significant correlation between pairs of cells (ρ > 0.3). The spread of calcium signals between cells results in detecting time delays in the increase in Ca^2+^ concentration. The maximum correlation between a pair of cells by a significant time shift between calcium signals can be explained by direct communication. In turn, this causality allows us to determine and assess the rate of propagation of Ca^2+^ waves in primary neuronal culture.

Five key parameters were used for the network analysis: the average level of correlation of all and adjacent cells, the average number of connections in a cell, the percentage of existing connections from the total number of possible connections, and the average propagation speed of delays between calcium signals.

### 4.8. RNA Extraction and RT-qPCR

Quantitative real-time PCR analysis was used to analyze the expression level of the transcription factor HIF-1α (Hif1), the neurotrophic factor BDNF, tropomyosin receptor kinase B (TrkB) and vitamin D receptor (VDR). Total RNA was isolated from primary neuronal cultures 24 h after the modeled hypoxia using an ExtractRNA kit (Evrogen, Moscow, Russia). Then, cDNA was synthesized using the Moloney murine leukemia virus (MMLV) reverse transcriptase (Evrogen, Moscow, Russia) and a random primer. 

Quantitative real-time PCR was performed using qPCRmix-HS SYBR + LowROX (Evrogen, Moscow, Russia) and an Applied Biosystems 7500 RT-PCR thermal cycler (Applied Biosystems, Foster City, CA, USA).
The following pairs of primers were used:Oaz1_fw5′-AAGGACAGTTTTGCAGCTCTCC-3′;Oaz1_rv5′-TCTGTCCTCACGGTTCTTGGG-3′;Hif1a_fw15′GCAATTCTCCAAGCCCTCCAAG-3′;Hif1a_rv15′-TTCATCAGTGGTGGCAGTTGTG-3′; mBDNF_fw 5′CCCAACGAAGAAAACCATAAGGA-3′;mBDNF_rv 5′-CCAGCAGAAAGAGTAGAGGAGGCT-3′;VDR_fw 5′-ATGCCCACCACAAGACCTACGAC-3′; VDR_rv 5′-AGT CTCCGGAGA AGCTGAGTGTGG-3′;TrkB_fw15′-TTTCCGCCACCTTGACTTGTCT-3′;TrkB_rv15′-GTCGGGGCTGGATTTAGTCTCC-3′;

The results were processed by the ΔΔCt method using a reference sample, in which the level of expression of the target genes was taken as a unit. Normalization was performed relative to the reference gene (Oaz1).

### 4.9. Statistical Analysis

Quantitative results are presented as the mean ± standard mean error (SEM) for normal distributions or as a median value and second and third interquartile range. Statistical analyses were performed using ANOVA implemented in Sigma Plot 11.0 software (Systat Software, Inc.). The Tukey post hoc test was used as a post hoc test following ANOVA. At least three independent biological replicates were used for all experiments. Differences between groups were considered significant if the corresponding *p*-value was less than 0.05.

## 5. Conclusions

In summary, our findings suggest that a critical increase in vitamin D3 can have a detrimental effect on the functional activity of brain neuron–glial networks, and become a trigger for irreversible catastrophic changes in hypoxic conditions. On the other hand, low concentrations of vitamin D3 have a pronounced neuroprotective effect. Further studies of the molecular–cellular mechanisms of vitamin D3 action on the neuron–glial networks functional activity could serve as the basis for the creation of new neuroprotectors and neuromodulators, and determine the optimal preventive and therapeutic drug dosages.

## Figures and Tables

**Figure 1 ijms-22-05417-f001:**
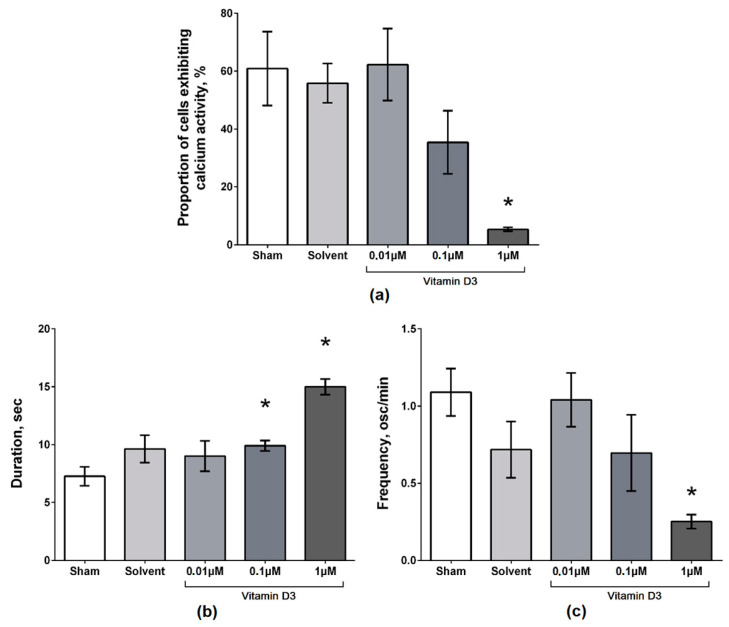
Main parameters of spontaneous calcium activity of primary neuronal cultures on day 7 after vitamin D3 application. (**a**) Proportion of cells exhibiting calcium activity; (**b**) Duration of Ca^2+^ oscillations, sec; (**c**) Frequency of Ca2+ oscillations (osc/min). * versus “Sham”, *p* < 0.05, the Mann–Whitney test.

**Figure 2 ijms-22-05417-f002:**
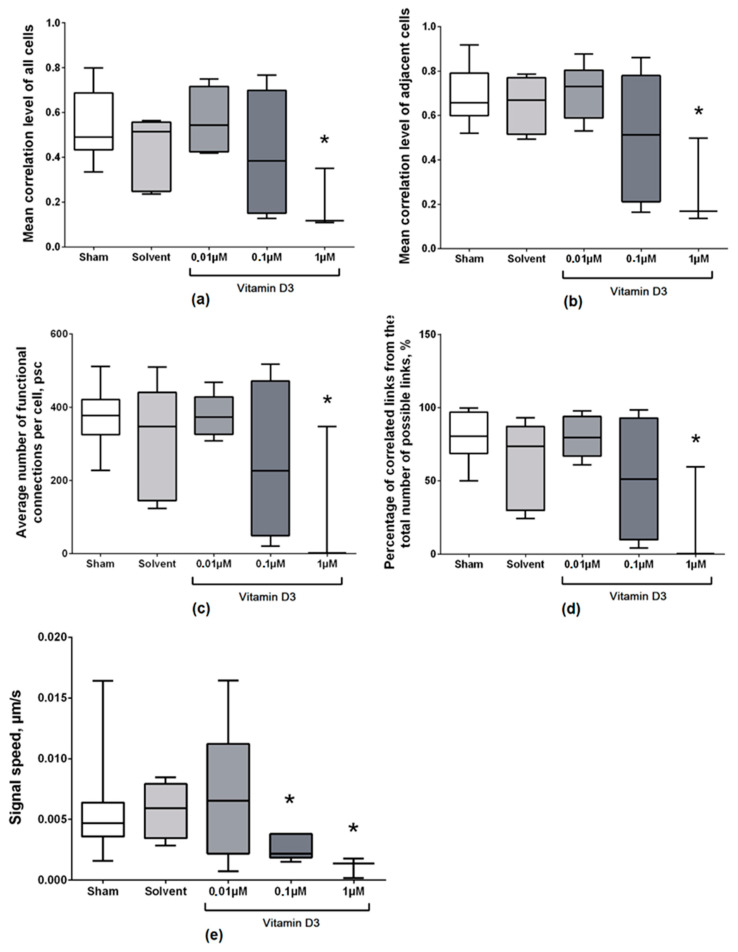
Main parameters of the network activity of primary neuronal cultures on day 7 after vitamin D3 application. (**a**) Mean correlation level of all cells in the culture; (**b**) Mean correlation level of adjacent cells; (**c**) Average number of functional connections per cell; (**d**) Percentage of correlated connections from the total number of possible connections; (**e**) Signal delay rate between cells. * versus “Sham”, *p* < 0.05, the Mann–Whitney test.

**Figure 3 ijms-22-05417-f003:**
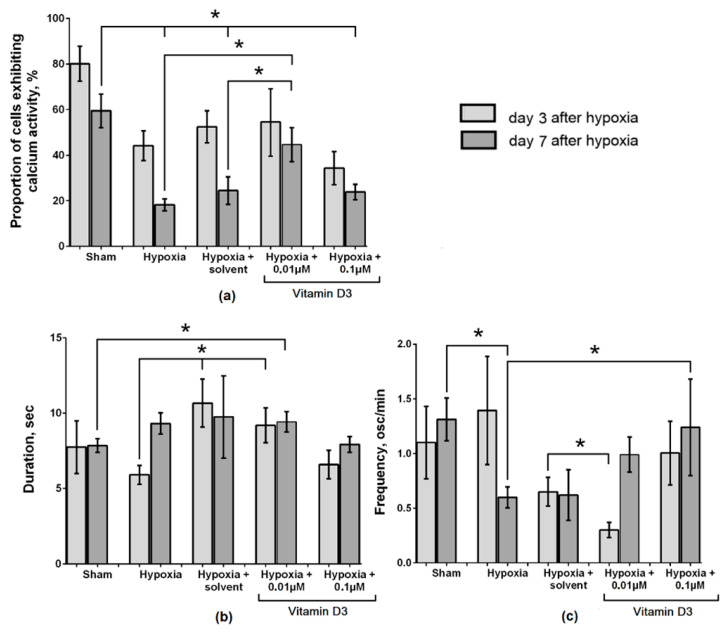
Main parameters of spontaneous calcium activity of primary neuronal cultures in the post-hypoxic period. (**a**) Proportion of cells exhibiting calcium activity; (**b**) Duration of Ca^2+^ oscillations, sec; (**c**) Frequency of Ca^2+^ oscillations (osc/min). * *p* < 0.05, one-way ANOVA and Tukey post hoc test.

**Figure 4 ijms-22-05417-f004:**
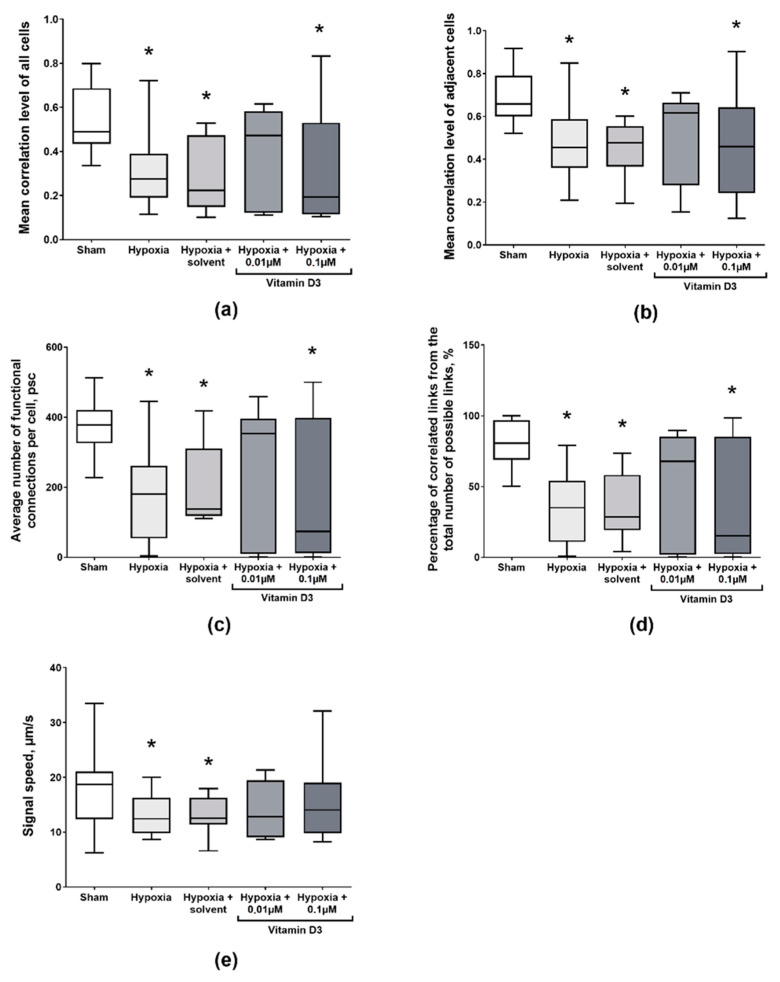
Main parameters of the network activity of primary neuronal cultures on day 7 of the post-hypoxic period. (**a**) Mean correlation level of all cells in the culture; (**b**) Mean correlation level of adjacent cells; (**c**) Average number of functional connections per cell; (**d**) Percentage of correlated connections from the total number of possible connections; (**e**) Signal delay rate between cells. * versus “Sham”, *p* < 0.05, the Mann–Whitney test.

**Figure 5 ijms-22-05417-f005:**
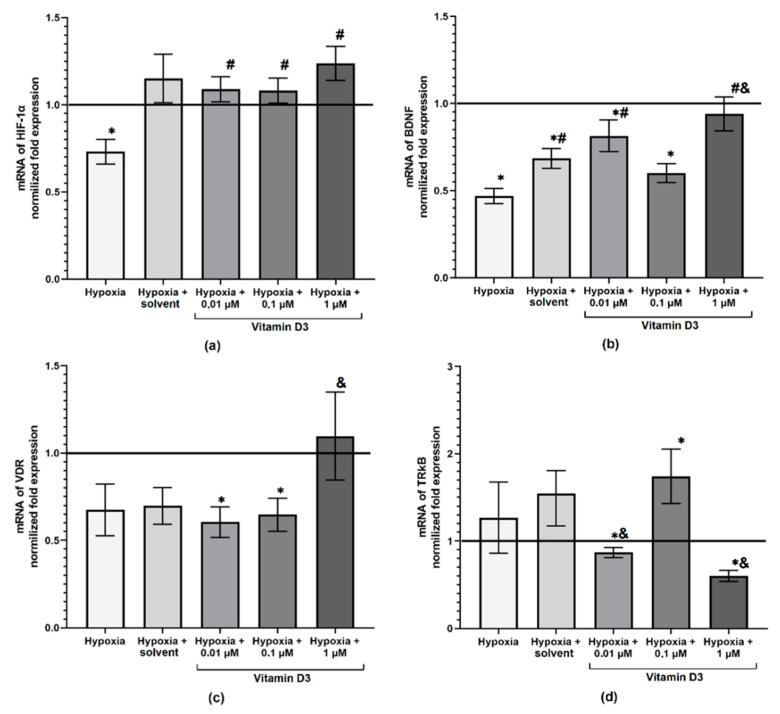
The level of transcription factor HIF-1α (**a**) and neurotrophic factor BDNF (**b**) genes expression in primary neuronal cultures cells 24 h after hypoxia modeling. Data are normalized to the reference gene (Oaz1), * versus “Sham”, # versus “Hypoxia”, & versus “Hypoxia + solvent”, *p* < 0.05, one-way ANOVA and Tukey post hoc test.

**Table 1 ijms-22-05417-t001:** Cell viability of primary neuronal cultures on day 7 after vitamin D3 application.

Group	Number of Viable Cells, %
Sham	93.91 ± 1.24
Solvent	89.34 ± 2.21
Vitamin D3 0.01 µM	89.99 ± 1.98
Vitamin D3 0.1 µM	91.97 ± 1.86
Vitamin D3 1 µM	88.64 ± 3.09

No statistical differences versus “Sham”, *p* < 0.05, one-way ANOVA and Tukey post hoc test.

**Table 2 ijms-22-05417-t002:** The effects of the use of vitamin D3 on cell viability of primary neuronal cultures in the post-hypoxic period.

Group	Number of Viable Cells in the Post-Hypoxic Period, %	
	Day 3	Day 7
Sham	96.88 ± 0.49	96.32 ± 0.53
Hypoxia	78.06 ± 2.82 *	80.39 ± 2.17 *
Hypoxia + solvent	81.52 ± 1.89 *	80.91 ± 2.37 *
Hypoxia + vitamin D3 0.01 µM	95.14 ± 0.7 *	89.23 ± 0.99 *#
Hypoxia + vitamin D3 0.1 µM	93.58 ± 1.1 *#	86.89 ± 0.71 *#
Hypoxia + vitamin D3 1 µM	8.89 ± 2.6 *#	-

* versus “Sham”, # versus “Hypoxia”, *p* < 0.05, one-way ANOVA and Tukey post hoc test.

## Data Availability

The data used to support the findings of this study are available from the corresponding author upon request.

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
