# Peer review of "Double-Edged Sword of Vitamin D3 Effects on Primary Neuronal Cultures in Hypoxic States"

_ijms, 2021, doi:10.3390/ijms22115417_

Round 1

Reviewer 1 Report

The manuscript “Neuroprotective Effect of Vitamin D3 on Primary Neuronal Cultures in Hypoxic States” aims to study a possible neuroprotective effect of vitamin D3 against hypoxic death in primary neurons. The topic is  interesting in neurobiological research, however there are some major concerns that need to be addressed.

  • Cell viability and death should be better characterized. In particular, experiments demonstrating the type of cell death that authors observe should be performed: is it apoptosis or necrosis or necroptosis or other? Furthermore, it is interesting to study whether there are more types of neuronal death, as the different neuroprotective/neurotoxic effect of vitamin D3 could also be linked to that.
  • The calcium activity in hypoxic cells and following vitamin D3 treatments should be better characterized too, in order to understand whether the observed differences are due to different membrane calcium channels activities or to the release/uptake from internal stores.
  • Considering the HIF1α and BDNF mRNAs expression, the effect of solvent (Figure 5) seems to be similar to the vitamin D effect, therefore other target genes should be tested and more careful controls should be performed.

Reviewer 2 Report

The study investigates the effects of vitamin D on the viability, calcium ion oscilations and expression levels of Hif-1a and BDNF in embryonic neuronal cells in cultures under normoxic and hypoxia-reoxygenation conditions. Authors show that Vit.D had no effect on neuronal viability under normoxic conditions but had some effects on spontanious calcium oscilations particularly at high 1 mM concentration. Under hypoxia/reoxygenation, lower concentrations of Vit.D had no significant effect on neuronal viability 3-5 days after insult, however 1 mM Vit.D killed neurons during 3-7 days post hypoxia. Effects of Vit.D on cellular functions are widely investigated. This paper could present some interesting insights into effescts of Vit.D on functioning of neuronal cells, calcium signaling and expression of some signaling molecules, however, there are important shortcomings in this study.

  1. Though authors tend to conclude that lower Vit.D concentrations had a protective effect against hypoxia-induced changes in calcium signaling, however, the effects seem to be negligible and do not pass a rigorous statistical examination: data presented in Figs. 3-4 show that Hypoxia plus Vit. D groups (both concentrations) were not significantly different from Hypoxia plus solvent group. Therefore, the effects may be related to application of ethanol rather than to Vit.D itself. The same seems to be true for data presented in Fig. 5.
  2. Ethanol by itself may strongly affect developing neurons. What concentrations of ethanol were used in the experiments and whether the same concentration of ethanol was added to cell cultures together with various Vit.D concentrations?
  3. Embryonic neuronal-glial cell cultures were used in this study. This may be some disadvantage as neurons may be premature and do not express, for example, glutamate receptors that are important in hypoxic/ischemic conditions. The composition of such cultures (neurons, astrocytes, microglia) are not described, and this is important as well because presence of glial cells may modulate neuronal responses to hypoxia. Does the percentage of glial cells change during 3-7 days reoxygenation?
  4. For viability assays, how many cells/field were calculated (or what was total number/condition of cells calculated in each experiment)?
  5. Table 2: about 5% difference in cell viability comparing Vit.D treated cells and treated with solvent seems biologically non-significant. How would authors explain an increase in percentage of viable cells after 7 days compared with 3 days?
  6. In the experiments of this study, cells were exposed to very short period of hypoxia – just 10 min. How the level of oxygen was controled? Also, it is not clear how sham control cells were treated in these experiments.

Reviewer 3 Report

The neuroprotective effect of vitamin D has previously been established. The current manuscript investigated primary neuronal culture to reveal the neuroprotective mechanisms. As established previously, low dose of vitamin D increased neuronal survival in the hypoxia model. The novelty of the findings stems from measuring calcium levels and functional connectivity between cells, as well as the death of neurons following high dose of vitamin D.

Major critiques

  1. The title suggest neuroprotective effects of vitamin D in primary neuronal culture, which does not include the much more dramatic degenerative effects of higher dose of vitamin D.
  2. A paper with the same conclusion has been recently published, which was not even discussed here: Cui et al, (2020) Vitamin D Attenuates Hypoxia-Induced Injury in Rat Primary Neuron Cells through Downregulation of the Dual Oxidase 1 (DUOX1) Gene. Med Sci Monit, 26: e925350.DOI: 10.12659/MSM.925350.
  3. For the hypoxia model, the description of the degree of hypoxia was not precisely reported (“a low oxygen content”)
  4. For the qRT-PCR study, it was not defined what Oaz1 is and why it can be used as a housekeeping gene instead of the usually applied 3-4 different housekeeping genes (GAPDH, actin, and so on).
  5. Statistics was described as “ANOVA”. It should be presented for each experiment why ANOVA can be used, if it is one or two-way ANOVA, is it repeated measure or not. What are F values, which post-hoc test was applied, etc.
  6. It would be nice to see an original image of video of Ca waves.
  7. The most serious critique of the manuscript is that the solvent was as effective as vitamin D to alter HIF and BDNF levels. How is it then possible to conclude the effect of vitamin D on the mRNA levels?

Minor point: VDR was not defined.

Round 2

Reviewer 1 Report

Authors performed all the experiments suggested and strongly improved the manuscript in all sections.

Reviewer 2 Report

Authors properly addressed questions and improved technical part of the study.

This manuscript is a resubmission of an earlier submission. The following is a list of the peer review reports and author responses from that submission.